# Development and Road Performance Verification of Aggregate Gradation for Large Stone Asphalt Mixture

**DOI:** 10.3390/ma17235712

**Published:** 2024-11-22

**Authors:** Yufeng Bi, Minghao Mu, Lujun Zeng, Tingting Ding, Chengduo Qian, Deshui Yu, Yingjun Jiang

**Affiliations:** 1Innovation Research Institute of Shandong Expressway Group Co., Ltd., Ji’nan 250000, China; zll0754@163.com (Y.B.); 2023221322@chd.edu.cn (M.M.); 2023221346@chd.edu.cn (C.Q.); 2Key Laboratory for Special Area Highway Engineering of Ministry of Education, Chang’an University, Xi’an 710064, China; 2023221336@chd.edu.cn; 3Shandong Provincial Communications Planing and Design Institute Group Co., Ltd., Ji’nan 250000, China; 2023221315@chd.edu.cn (T.D.); 18870380957@163.com (D.Y.)

**Keywords:** road engineering, large stone asphalt mixture, strong interlocking densely graded structure, mechanical strength, road performance

## Abstract

The pavement base and subbase are the main load-bearing structures of asphalt pavement, and their materials need to have sufficient bearing capacity. Therefore, in the development of LSAM-50 mixtures with higher bearing capacity, after significant research and engineering practice, conventional particle size asphalt mixtures have formed their own excellent mineral gradation and have been incorporated into relevant specifications, while LSAM-50 mixtures, including mineral gradation, have not been involved in related research and engineering applications. According to the strength composition mechanism of asphalt mixtures, under the same circumstances of asphalt, due to the large nominal maximum particle size of LSAM-50 and the small amount of asphalt used, the strength of mineral grading is more important than that of asphalt, which is one of the key issues to be solved in the research of LSAM-50 mixtures. This study aims to enhance the road performance of asphalt mixtures with a maximum nominal particle size of 50 mm (LSAM-50). The variation of void ratios in coarse aggregate skeletons was investigated when aggregates of 37.5–53 mm (designated as D1), 19–37.5 mm (designated as D2), and 9.5–19 mm (designated as D3) were mixed in different proportions. Meanwhile, the effects of fine aggregate gradation on the strength of asphalt mortar and the influence of the ratio of coarse to fine aggregates on the mechanical strength of LSAM-50 were examined. A densely graded structure with strong interlocking for LSAM-50 was proposed, and its road performance was verified. The results indicate that when the ratio of D1, D2, and D3 is 5:2:3, the void ratio of the mixed coarse aggregate is minimized. When the decrement factor i is 0.75, the compressive strength and splitting strength of asphalt mortar reach their maxima. Compared with the densely graded asphalt-stabilized aggregate mixture (ATB-30) with a maximum nominal particle size of 37.5 mm, the dynamic stability of LSAM-50 with the proposed gradation is increased by 400%, the low-temperature bending strain by 3%, the SCB bending strength by 47%, and the residual SCB strength by 90%.

## 1. Introduction

In recent years, there has been a growing interest in developing large stone asphalt mixtures (LSAM) to enhance road performance, particularly for heavy-duty pavements [1,2,3]. Large stone asphalt mixtures with a nominal maximum aggregate size of 50 mm (LSAM-50) have the potential to improve load-bearing capacity and reduce deformation under traffic loads [4]. The continuous increase in traffic volume has placed higher demands on the pavement performance of asphalt mixtures [5]. Gradation is one of the main factors affecting the mechanical strength and durability of asphalt mixtures [6,7,8,9]. Consequently, numerous studies have been conducted by road researchers to explore the gradation of asphalt mixtures.

Gürer et al. studied the impact of gradation on the performance of stone mastic asphalt (SMA) mixtures and found that the use of mixed aggregates, maximum aggregate size, and percentage of coarse aggregate, along with carbon fibers, had a positive effect on SMA performance [10]. Wang et al. studied the sensitivity of the Marshall index of different gradation mixtures to the asphalt-aggregate ratio. They found that the effective asphalt-aggregate ratio range varies for different gradation mixtures. By analyzing the fractal dimension values, the performance of the mixtures can be preliminarily predicted [11]. Guo et al. proposed a gradation optimization method by studying the microscopic mechanism of how particles of different sizes in asphalt mixtures are stressed during rutting formation [12]. Lira et al. developed a gradation analysis framework based on packing theory to evaluate the rutting performance of asphalt mixtures based on aggregate gradation [7,13]. Ahmad et al. designed the gradation of asphalt mixtures using both traditional methods and the Bailey method. They found that mixtures designed using the Bailey method had lower permeability. Additionally, as the nominal maximum aggregate size increased, the permeability of the mixtures also increased [14]. Xue et al. studied the effect of the composition of various dispersed systems on the performance of stone mastic asphalt (SMA-13) mixtures with a nominal maximum aggregate size of 13.2 mm based on mortar theory and proposed the optimal gradation composition for coarse aggregates [15]. Li et al. studied the impact of gradation parameters on the performance of asphalt rubber stress-absorbing membrane interlayers and found that gradation is a key factor influencing the stress-absorbing performance of the mixtures [16]. Shi et al. proposed a method using a two-parameter Weibull distribution model to describe aggregate gradation. They studied the impact of gradation characteristic parameters on the volumetric indices of the mixture and found a linear relationship between the gradation characteristic parameters and the mixture’s volumetric indices [17]. Yan et al. found that fine aggregates play a crucial role in determining the cracking performance of asphalt mixtures [18]. Xiong et al. studied the relationship between the volumetric characteristics of skeleton gradation and the mechanical properties of semi-flexible pavement materials [19]. Kosparmakova et al. studied the impact of aggregate gradation on the durability of asphalt mixtures. The results showed that, under hot climatic conditions, asphalt mixtures with gradations higher than +4% and 2% of the upper and lower standard values exhibited the strongest resistance to water damage and the least rutting [20]. Feng et al. found that the uniformity of large stone asphalt mixtures deteriorates as the nominal maximum aggregate size increases [3]. Chen et al. studied the effects of different roughness and particle sizes on the interlocking and contact forces of coarse aggregates. They proposed an optimal ratio of coarse to fine aggregates that can improve the dynamic stability of asphalt mixtures by 30% and the fracture toughness by 26% [21]. Ji et al. found that the nominal maximum aggregate size significantly affects the permanent deformation resistance, low-temperature cracking resistance, and moisture sensitivity of asphalt mixtures. Larger aggregate sizes enhance the rutting resistance of the mixtures. Based on numerical simulation methods, they developed a gradation optimization method for mixtures, which can improve the shear strength and dynamic stability of the mixtures by 25.5% and 27.0%, respectively [2,22]. Zhang et al. studied the impact of reclaimed asphalt pavement (RAP) gradation on the performance of recycled asphalt mixtures. The results showed that the finer the RAP gradation, the poorer the mixture’s water stability and thermal cracking performance. It is recommended to avoid using fine RAP particles in cold and rainy regions [23]. Zhang et al. used a microscopic approach to reveal the impact of gradation design on the performance of CMEAM pavements [24].

The aforementioned researchers have conducted extensive studies on the gradation of asphalt mixtures, but most of their work focuses on mixtures with a nominal maximum aggregate size of ≤37.5 mm. While these studies provide valuable insights, their findings cannot be directly applied to the gradation design of LSAM-50 mixtures. Research on the optimal gradation design for LSAM-50 is still insufficient. The inclusion of coarse aggregates with a nominal maximum size of 53 mm in LSAM-50 mixtures changes the composition of the aggregates. Traditional asphalt mixtures typically use smaller aggregates, and their gradation principles may not be directly applicable to larger particles. The interlocking force and void ratio within the coarse aggregate skeleton play a crucial role in the mechanical strength and durability of the mixture [25,26]. Therefore, it is necessary to study the optimal proportions of larger and more varied-sized coarse aggregates to improve the road performance of LSAM-50.

This study aims to develop and validate the gradation of LSAM-50, focusing on the proportionate mixing of 37.5–53 mm, 19–37.5 mm, and 9.5–19 mm aggregates. By analyzing the void ratio changes in coarse aggregate skeletons and the effects of fine aggregate gradation on asphalt mortar strength, the study seeks to establish an optimal gradation that maximizes the mechanical strength and road performance of LSAM-50. The proposed gradation is then compared with conventional dense-graded asphalt stabilized aggregate mixtures (ATB-30) to verify its superiority in terms of dynamic stability, low-temperature bending strain, SCB bending strength, and residual SCB strength. The findings of this research will contribute to the development of high-performance asphalt mixtures for heavy-duty pavements, offering significant improvements in durability and load-bearing capacity.

## 2. Materials and Methods

### 2.1. Materials

#### 2.1.1. Asphalt

Esso A-70 base asphalt produced in Singapore was used in all types of asphalt mixtures that were studied in this article; the technical properties are shown in Table 1.

#### 2.1.2. Aggregates

In this study, limestone was used as the aggregate, which was produced in Baoji, Shaanxi, China. The technical properties are listed in Table 2.

### 2.2. Methodology

Through laboratory compaction tests, the optimal composition of coarse aggregates was proposed based on the principle of minimizing the interstitial void ratio. The i method was used to determine the proportion of fine aggregates in asphalt mortar. Fine aggregate gradations were calculated with different *i* values (0.65, 0.7, 0.75, 0.8) and asphalt mortar specimens were formed. The optimal i value corresponding to the fine aggregate gradation was determined based on the highest unconfined compressive strength and splitting strength of the asphalt mortar specimens. By filling the voids in the coarse aggregate with the mortar, different gradations were simulated to identify the range of gradations with the best mechanical properties. Finally, the road performance of the LSAM-50 gradation proposed in this paper was validated.

### 2.3. Test Methods

#### 2.3.1. Specimen Preparation Test

In this study, the asphalt mixture cylindrical specimens were molded using vertical vibration test equipment (VVTE). The specimen dimensions were 200 mm in diameter and 160 mm in height; the VVTE working parameters are shown in Table 3.

#### 2.3.2. Mechanical Strength Test

The unconfined compressive strength (*R*_c_) was tested according to T0713-2000 in the “Test Methods of Bitumen and Bituminous Mixtures for Highway Engineering” (JTG E20-2011) [27]. The specimens were placed in a constant temperature air box at 60 °C for 8 h before testing the *R*_c_ using a universal testing machine. The loading rate during the test was 2 mm/min. The *R*_c_ test of the LSAM-50 mixture specimens is shown in Figure 1.

The *R*_c_ of the LSAM-50 mixture specimens was calculated according to Equation (1).
(1)$$R_{c}=\frac{4P}{\pi \phi^{2}},$$
where *R*_c_ represents the unconfined compressive strength (MPa), *P* represents the peak load (N), and *Φ* represents the specimen diameter (mm).

The splitting test (*R*_T_) is shown in Figure 2a. The *R*_T_ was conducted according to T0716-2011 in the “Test Methods of Bitumen and Bituminous Mixtures for Highway Engineering” (JTG E20-2011). The specimens were placed in an oven at 60 °C for 8 h before testing the *R*_T_ using a universal testing machine. During the test, the loading rate of the materials testing machine was 50 mm/min, using 25.4 mm circular arc loading strips. The corresponding indirect tensile test fixtures were also prepared, as shown in Figure 2b,c.

The formula for calculating the *R*_T_ of LSAM-50 mixture specimens, derived based on elasticity theory, is shown in Equation (2).
(2)$$R_{i}=\frac{2P}{\pi ah}\left(\sin2\alpha -\frac{a}{b}\right),$$
where *R*_T_ represents the splitting strength (MPa), *P* represents the peak load (N), *a* represents the width of the loading strip (mm), *d* represents the specimen diameter (mm), *h* represents the specimen height (mm), and *α* represents the central angle corresponding to half the width of the loading strip (°).

The semi-circular bending strength test (SCB), as a non-standard test, is used in this study solely to evaluate the low-temperature performance of the mixture based on its strength. The test does not involve the calculation and use of fracture energy, toughness, or stiffness. The test is conducted by applying a concentrated load at the midpoint of a simply supported semicircular specimen until failure. The tensile strength of the material is calculated from the maximum load at failure, as shown in Equation (3). The test procedure is depicted in Figure 3.
(3)$$\sigma =\frac{12Pa}{t\phi^{2}},$$
where *σ* represents the semi-circular bending strength (MPa), *P* represents the ultimate load (N), *a* represents half the span between supports (mm), generally *a* = 0.4*Φ*, *t* represents the specimen thickness (mm).

## 3. Mechanism of Strength Formation and Gradation Design Method

### 3.1. Mechanism of Strength Formation

Asphalt mixture is a composite material composed of asphalt, coarse aggregate, fine aggregate, mineral powder, and additives. From the perspective of gel theory, asphalt mixture is a dispersed system characterized by a multistage spatial network gel structure.

The gradation design of the LSAM-50 mixture should have sufficient coarse aggregate to interlock and form a skeleton, as well as sufficient “dispersed medium” to fill the voids and form a dense skeleton structure. In this context, the strength of the LSAM-50 mixture primarily originates from the internal friction formed by the interlocking coarse aggregates and the cohesion of the asphalt mortar. The cohesion of the asphalt mortar mainly depends on the gradation of fine aggregates, the properties of the asphalt, and the asphalt content, whereas the interlocking force of the coarse aggregate mainly depends on the properties of the coarse aggregate and its proportion.

### 3.2. Gradation Design Method

The basic principle of the gradation design for the LSAM-50 mixture is that the coarse aggregate forms a skeleton, with the asphalt mortar densely filling the skeleton formed by the coarse aggregate. At the same time, it is crucial to ensure that the fine aggregate does not disrupt the coarse aggregate skeleton. The gradation design concept mainly includes the following parts:(1)Calculate the particle size relationship between primary and secondary fillers based on the ordered packing model of spherical particles, and reasonably grade the LSAM-50 coarse aggregate in combination with the commonly used coarse aggregate gradations in actual engineering.(2)Guided by the existing research results of particle packing theory, conduct a study on the reasonable proportioning of coarse aggregates. Optimize the proportion of coarse aggregates based on the principle of optimal void ratio.(3)Calculate the fine aggregate gradation using the i-method formula and form large-size specimens with dimensions of Φ200 mm × h160 mm. Optimize the fine aggregate gradation based on the principle of optimal mechanical strength.(4)Study the effect of the proportion of coarse aggregate to asphalt mortar on the mechanical properties of the LSAM-50 mixture and propose a densely graded, strong interlocking skeleton with the optimal mechanical properties for the LSAM-50 mixture.

## 4. Development of a Densely Graded Strong Interlocking Skeleton for LSAM-50

### 4.1. Theoretical Research and Gradation of Coarse Aggregate

#### 4.1.1. Theoretical Research

The stacking theory was proposed in 1940 and is widely used to determine whether the porosity of road materials meets the requirements of use. When applied to actual production, it will be supplemented by corrections according to the shape of the material particles, forming pressure and use requirements. For asphalt mixtures, the essence of the forming process is to make the aggregate particles fill the mold space as evenly and densely as possible. To form a compact structure of LSAM-50, a strong, embedded, and extruded skeleton, that is, to solve the problem of how the aggregate particles are tightly packed. The classical particle stacking theory was mainly proposed by Fuller, Horsfield, Dinger, and Funk, and from this, the theory of continuous-size particles and discontinuous-size particles was derived. The researchers’ exploration of the particle stacking problem, including theoretical calculations and laboratory tests, reached a conclusion that has practical significance for the molding process:(1)Single-particle-sized particles cannot achieve a tight packing state, and multiple particle sizes can be used to achieve a tight packing state. The larger the difference in particle size between different particle sizes, the tighter the packing, and the difference is usually more than 4 to 5 times.(2)The number of smaller particles should be able to fill the gap between the particles and the larger particles, and the number is related to the appearance and filling method of the particles and should not interfere with the larger particles.(3)Adding components with different particle sizes can increase the bulk density, but the actual significance is small when the components are greater than 3. When there are two components, the ratio of coarse and fine particles should be 7:3; when there are three components, the ratio of coarse and fine particles should be 7:1:2.

#### 4.1.2. Gap Ratio of Stacking Theory

The packing theory primarily focuses on the ways in which granular substances (such as sand and powder) are packed in space. Moreover, its packing models can also be applied to aggregates. Granular packing can be categorized into ordered and disordered types. Ordered packing usually encompasses cubic close packing, hexagonal close packing, complex hexagonal packing, and pyramidal packing, all of which exhibit relatively high packing efficiencies. On the other hand, disordered packing is more haphazard, just like the ordinary sand piles we encounter in daily life, where the arrangement of particles lacks a fixed pattern. The porosity is defined as the ratio of the void volume among particles within the particle packing body to the total volume. It is a crucial parameter for describing the compactness of the packing structure.

(1)single particle size particle accumulation

In the accumulation theory, the accumulation mode of single-size particles is generally shown in Figure 4, and the clearance ratio is shown in Table 4.

However, under realistic stacking conditions, studies have found that the most likely random stacking gap rate is 37.5% [28].

(2)Aggregates of various specifications

The particle accumulation model is mainly based on discontinuous particle size. The representative Furnas theory holds that the closest accumulation is formed when small particles just fill the pores of large particles [29,30]. When there are three sizes of particles, the middle particle size should just fill the gap of the largest particle size, and the smallest particle size should fill the gap of the larger particle, which can be extended to the case of multiple size particles.

Based on the accumulation theory of discontinuous-sized particles, Westman and Hugill [31] calculate the maximum accumulation factor of multi-sized particles and also enumerate the calculation steps of the mixture of 2 and 3 sizes of particles. For the continuous particle accumulation particle size, the advocate Andreasen describes the particle distribution as a “statistically similar” distribution form and proposes a model equation, which believes that to minimize the porosity, the distribution modulus should be 0.33~0.55. However, the actual particle is different from the ideal sphere, so Dinger and Funk modify the model equation proposed by Andreasen, introduce the finite small minimum particle size in the distribution, and propose the Dinger-Funk equation, which holds that the continuous distribution sphere has the maximum density and the smallest porosity when the distribution modulus is 0.37. According to existing studies, the porosity of refractory materials stacked at 3 to 5 mm and 1 to 3 mm is shown in Table 5.

#### 4.1.3. Theoretical Research on the Classification of Coarse Aggregate

The spatial distribution of coarse aggregate in the mold is difficult to accurately grasp due to the influence of its geometric shape and mechanical properties. To simplify the study, aggregates of different sizes are approximated as spheres with different diameters, simulating their packing state in the mold. The ordered packing state of single-sized particles in space can be simplified into a regular polyhedron model with spherical particles as the vertices. Figure 5 shows the ball-and-stick models of the spatial distribution of spherical particles under several binary component states.

Unlike other ordered packing models, the binary component ordered packing model of the hexahedron exists in only one form, and its front view, side view, and top view are the same in two-dimensional representation. From the ball-and-stick model, it can be seen that the particle center distance in the ordered packing state of the hexahedron is the largest. This means that the packing model in this state is the most open and ready to be filled. Using this model to simulate the particle packing state in a limited space specimen is more conservative and stable for particle gradation.

In the ordered packing model of the hexahedron, the eight spherical particles at the vertices form a regular hexahedral void. Through simple geometric calculations, it can be determined that the maximum radius of small particles that can fill this void is 0.732 times the radius of the large particles at the vertices. The small particles fill the center of the hexahedral void and contact the eight large particles. The four large particles at the same layer vertices form a plane and interact with the small particles in four directions. Under the combined action of these contact forces and gravity, the small particles can reach a balanced state and stably embed into the hexahedral void without disrupting the ordered skeleton formed by the large particles.

The maximum particle size of the LSAM-50 mixture aggregate is 37.5–53 mm. To prevent the primary aggregate skeleton from being disrupted, the conservative calculation for the particle size of the secondary filler should be 19–26.5 mm aggregate, resulting in a gap-graded, coarse aggregate gradation for the LSAM-50 mixture. Conversely, calculating the secondary filler particle size with an average of 45 mm yields 31.5–37.5 mm aggregate, resulting in a continuous coarse aggregate gradation for the LSAM-50 mixture. The project formed VVTM specimens of gap-graded and continuous-graded mixtures with optimal void ratios for coarse aggregate packing and tested their mechanical strengths. The results showed that the mechanical strength of the gap-graded mixture was less than 70% of that of the continuous-graded mixture.

The filler particle size calculated from the ordered packing model only sets an upper limit, while the lower limit is considered to have minimal impact on the skeleton and is not explicitly restricted. Based on the indoor test results and commonly used aggregate gradations in actual engineering, the recommended aggregate gradation for the LSAM-50 mixture is as follows: 37.5–53 mm aggregate (designated as D_1_), 19–37.5 mm aggregate (designated as D_2_), 9.5–19 mm aggregate (designated as D_3_), 4.75–9.5 mm aggregate (designated as D_4_), 0–4.75 mm aggregate (designated as D_5_). Initially, D_2_ is subdivided into 19–26.5 mm, 26.5–31.5 mm, and 31.5–37.5 mm aggregates, each accounting for one-third. D_3_ is subdivided into 9.5–13.2 mm, 13.2–16 mm, and 16–19 mm aggregates, each accounting for one-third.

#### 4.1.4. Gradation of Coarse Aggregate

The gradation of coarse aggregate was determined using a proportional filling method, optimizing the aggregate ratio based on bulk density and void ratio. To ensure the filling results closely matched the actual conditions of the specimen, single-spec, double-spec, and triple-spec filling tests were conducted using a mold with dimensions of *Φ*200 mm × *h*200 mm. During the test, aggregates of each grade were weighed proportionally, filled into the mold, and leveled at the top. A spacer block was placed on top of the aggregates, and the height from the top of the mold to the spacer block was measured to calculate the aggregate packing height. Each mass ratio was tested once, with the aggregates emptied, remixed, and filled into an iron bucket for six tests to obtain an average. The testing method is shown in Figure 6.

During the test, the voids in D_1_ were filled by D_2_ particles, and the combined voids of D_1_ + D_2_ were filled by D_3_ particles. It should be noted that D_4_ should theoretically act as the third level of filling particles in the three-level filling test of coarse aggregates. However, due to the large particle size difference, D_4_ particles tend to concentrate at the bottom of the mold and fail to function effectively as fillers. Therefore, D_4_ and asphalt mortar should be combined as filler material to form specimens and study their optimal proportions. In this context, 4.75–9.5 mm aggregate is considered as the filler material for gradation research. This approach forms an important basis for the gradation research using two filling methods in this chapter. To prevent secondary particles from disrupting the skeleton structure, the secondary particles should not be larger than the size of the voids. Thus, a well-graded aggregate with both interlocking force and dense packing is achieved, characterized by strong interlocking skeleton gradation with both frictional resistance and cohesion. The calculation of the voids content (VCA) is shown in Equation (4), and the results of the filling tests are shown in Table 4, Table 5 and Table 6.
(4)$$VCA=1-\frac{\rho_{d}}{\sum_{1}^{i}\rho_{i}\times \omega_{i}},$$
where *ρ*_i_ represents the density of D_i_-grade aggregate (g/cm^3^), and *ω*_i_ represents the mass fraction of Di-grade aggregate (%).

As shown in Table 6, the actual void ratio of single-grade aggregates is greater than the theoretical void ratio due to the irregular shape and multiple edges of the aggregates. In contrast, the void ratio of mixed aggregates with continuous gradation is less than the theoretical void ratio of single-grade aggregates. The smaller the particle size, the smaller the packing void ratio. This is because as the particle size decreases, the difference between the particle size and the mold size becomes larger, reducing the boundary effect and making the void ratio closer to the theoretical value. For aggregates in the 37.5–53 mm range, the size of the mold limits the packing, resulting in a significant boundary effect. Additionally, the uneven particle size distribution and irregular shape of the aggregates cause the actual void ratio to be greater than the theoretical value.

As shown in Table 7 and Table 8, the void ratio decreases first and then increases as the proportion of secondary particle size aggregates increases. The test results are consistent with the theory, with the minimum void ratio achieved at a mass ratio of 70:30. The secondary particle size aggregates initially fill the voids in the larger particle size aggregates. However, as the voids become fully filled, the secondary aggregates begin to disrupt the skeleton structure of the larger aggregates. Thus, further increasing the proportion of secondary particle size aggregates does not achieve a dense packing state.

Based on the above test results, the following parameters are proposed for this stage of the research:(1)Single-grade D_1_ void ratio: 49.2%,(2)When D_1_:D_2_ = 7:3, the void ratio is 45.1%,(3)When D_1_:D_2_:D_3_ = 5:2:3, the void ratio is 44.3%.

### 4.2. Gradation of Fine Aggregate

The gradation of fine aggregate was calculated using the i-method, as shown in Equations (5) and (6).
(5)$$P_{x}=100{\left(i\right)}^{x}$$
(6)$$x=3.32\mathrm{lg}(\frac{D}{d_{x}})$$
where *P*_x_ represents the passing percentage of a certain aggregate size (%), *i* represents the decrement coefficient of the passing rate, *D* represents the nominal maximum size of the aggregate (mm), and *d*_x_ represents the particle size of a certain aggregate (mm).

Four sets of *i* values, namely 0.55, 0.65, 0.75, and 0.85, were selected to calculate the passing rates of fine aggregate for particle sizes below 4.75 mm. The gradations of fine aggregate corresponding to different i values are shown in Table 9.

The optimal fine aggregate gradation was selected based on the 60 °C compressive strength (*R*_c_) and 60 °C splitting strength (*R*_T_). The strength test results for each fine aggregate gradation at their respective optimal asphalt-aggregate ratios are shown in Table 10. The effects of the *i* values on the *R*_c_ and *R*_T_ of the asphalt mortar are illustrated in Figure 7. During the tests, the optimal asphalt-aggregate ratios for each fine aggregate gradation were as follows: 7.5% for *i* values of 0.55 and 0.65; 8.5% for an *i* value of 0.75; and 10% for an *i* value of 0.85.

As depicted in Table 11 and Figure 7, the peak strength of the asphalt mortar emerges at i = 0.75, and the corresponding fine aggregate gradation is presented in Table 9. The measured density of the asphalt mortar is 2.410 g/cm^3^. It can be observed from Figure 8 that the variation trend of compressive strength with respect to mortar increment approximates that of gross bulk density with mortar increment, reaching its peak when the mortar increment is 3%. The underlying reason is as follows: with the augmentation of the oil film thickness, the internal cohesion of asphalt within the asphalt mixture increases, thereby enabling a higher compressive strength to be achieved. When the filled asphalt mortar exceeds a certain quantity, the excessive asphalt generated will thicken the asphalt film, which is detrimental to the mutual extrusion among coarse aggregates. Consequently, when the mortar dosage surpasses the theoretical mortar dosage by +3%, a downward trend is exhibited. The cracking strength, on the other hand, tends to increase linearly. This is because increasing the proportion of mortar indirectly enhances the cohesion of the asphalt mixture, and the cracking strength is predominantly derived from cohesion, thus manifesting an approximately linear increase. Specifically, increasing the proportion of mortar implies an increase in the oil-stone ratio. Within a specific range of the oil-stone ratio, an increase in this ratio will improve the cohesion of the asphalt mixture, and since the splitting strength is largely sourced from cohesion, the splitting strength of the LSAM -50 mixture will rise with the increase in the proportion of mortar.

### 4.3. The Optimization of Coarse and Fine Aggregates

The optimization study of the coarse and fine aggregate proportions was conducted using the following two methods:

Method 1: Fine aggregate (0–4.75 mm, D_5_) asphalt mortar was used to fill coarse aggregate alone. The void ratios for the single-grade D_1_ (denoted as S grading) at 49.2%, dual-grade D_1_ + D_2_ (denoted as D grading) with the optimal condition D_1_:D_2_ = 70:30 at 45.1%, and tri-grade D_1_ + D_2_ + D_3_ (denoted as T grading) with the optimal condition (D_1_ + D_2_):D_3_ = 70:30 at 44.3% were considered as the theoretical void ratios for mortar filling. The fine aggregate grading was the optimal *i* method of grading, as shown in Table 12, with an optimal asphalt-to-stone ratio of 8.5%. Filling of coarse aggregate gradings S, D, and T was performed using theoretical amounts of asphalt mortar, theoretical amounts plus 3%, and theoretical amounts plus 6%. The theoretical amounts of mortar for different grading types are shown in Table 12. Evaluation was based on the physical properties of bulk volume density and void ratio, as well as the mechanical properties of compressive strength and splitting strength.

Method 2: A combination of fine aggregate (0–4.75 mm, D_5_) asphalt mortar and 4.75–9.5 mm aggregate (D_4_) was used to fill coarse aggregate. The ratios of D_5_ to D_4_ were 2:1, 3:1, and 4:1, and filling of coarse aggregate gradings was performed using theoretical amounts of asphalt mortar, theoretical amounts plus 3%, and theoretical amounts plus 6%. The theoretical amounts of mortar for different grading types are shown in Table 13. Evaluation was similarly based on the physical properties of bulk volume density and void ratio, as well as the mechanical properties of compressive strength and splitting strength.

#### 4.3.1. D_5_ Asphalt Mortar Was Used to Fill Independently

When D_5_ asphalt mortar is used for independent filling, the effects of asphalt mortar content and coarse aggregate grading type on the physical properties of LSAM-50 specimens are shown in Figure 8.

As shown in Figure 8, when the asphalt mortar is used to independently fill the three types of coarse aggregate gradings, the bulk volume density initially increases and then decreases with the increase in mortar content. The bulk volume density of each grading reaches its peak at the theoretical mortar amount plus 3%. This is because, as the asphalt mortar content increases, the voids are gradually filled until excess asphalt is produced, causing the bulk volume density to start decreasing. The void ratio shows an approximately linear decreasing trend with the increase in mortar content. The voids in the LSAM-50 mixture become more densely filled with mortar under compaction, and the excess asphalt further fills the voids.

Figure 8 also shows that when the three types of coarse aggregate gradings are filled with the theoretical amount of asphalt mortar, the theoretical amount plus 3%, and the theoretical amount plus 6%, the increase in coarse aggregate content leads to an increase in bulk volume density and a decrease in void ratio. This is because the minimum skeletal void ratio is achieved without interference, and as the aggregate content increases, the skeleton is gradually filled with secondary-sized aggregates and asphalt mortar.

The effects of asphalt mortar content and coarse aggregate grading type on the mechanical strength of LSAM-50 specimens are shown in Figure 9.

As shown in Figure 9, *R*_c_ follows a trend similar to that of the bulk volume density, reaching its peak at the theoretical mortar amount plus 3%. This is because as the coarse aggregates are gradually coated, the internal cohesion of the asphalt mixture increases, resulting in higher compressive strength. However, when the filled asphalt mortar exceeds a certain amount, the excess asphalt increases the thickness of the asphalt film, which is detrimental to the interlocking of the coarse aggregates. *R*_T_ shows an approximately linear increase with the increase in mortar content. This is because the increased proportion of mortar indirectly enhances the cohesion of the asphalt mixture, and the tensile strength is primarily provided by this cohesion.

Figure 9 also shows the impact of grading types on mechanical strength. When the three types of coarse aggregate gradings are filled with the theoretical amount of asphalt mortar, the theoretical amount plus 3% and the theoretical amount plus 6%, both *R*_c_ and *R*_T_ increase. This may be because, as the coarse aggregate content increases, the mixture gradually forms a higher interlocking force within the aggregate skeleton. Consequently, the coarse aggregate skeleton grading formed by D_2_ + D_3_ filling D_1_ (grading T) shows significantly better interlocking than D_2_ filling D_1_ alone (grading S). Therefore, the higher the interlocking force within the coarse aggregate skeleton, the higher the mechanical strength of the LSAM-50 mixture.

#### 4.3.2. D_5_ Asphalt Mortar Combined with D_4_ for Joint Filling

When D_5_ asphalt mortar is combined with D_4_ to fill different graded coarse aggregates, the effects of mortar increment, coarse aggregate grading, and the D_5_ to D_4_ ratio on the physical and mechanical properties of LSAM-50 specimens are shown in Table 14, Table 15, Table 16 and Table 17.

From Table 14, Table 15, Table 16 and Table 17, the following phenomena can be noted:(1)As the mortar content increases, the bulk volume density of LSAM-50 ascends, while the void ratio descends. The compressive strength (Rc) first rises and then falls, with most of the peak values emerging at a mortar increment of 3%. The tensile strength (RT) presents an upward trend as the mortar content increases. However, when the ratio of D_5_ to D_4_ increases to 4:1, RT first increases and then decreases with the growth of mortar content.(2)With the increase in coarse aggregate content, under each mortar content condition, the bulk volume density of LSAM-50 continuously decreases, whereas the void ratio first increases and then decreases. Under the three mortar increments and the three D_5_ to D_4_ ratios, for different mortar increments, the compressive strength (*R*_c_) and tensile strength (RT) of LSAM-50 specimens follow the order: T grading > D grading > S grading.

Based on Table 16 and Table 17, Figure 10, Figure 11, Figure 12, Figure 13, Figure 14 and Figure 15 illustrate the impacts of mortar increment, coarse aggregate grading, and the D_5_ to D_4_ ratio on the mechanical strength of LSAM-50 specimens.

As depicted in Figure 10, Figure 11, Figure 12, Figure 13, Figure 14 and Figure 15, within the S grading context, when the mortar increment ranges from 0% to 3%, the compressive strength (*R*_c_) ascends with an increase in the D_5_ to D_4_ ratio. Nevertheless, when the mortar increment reaches 6%, *R*_c_ first increases and then decreases, and the trend of compressive strength becomes relatively flat. The underlying reason is that prior to the mortar increment attaining 6%, the overall mortar content within the mixture is inadequate. Consequently, an elevated proportion of D_5_ aggregate is required to reinforce the filling effect of the fine aggregate on the voids of the coarse aggregate. For the D grading and T grading, Rc exhibits a tendency of first rising and then falling. This can be attributed to the high mortar content when the D_5_ to D_4_ ratio is 4:1. Under such circumstances, the free asphalt in the mixture overly lubricates the skeleton, thereby resulting in a reduction in mechanical strength. Concerning the tensile strength (*R*_T_), under theoretical mortar conditions, the *R*_T_ of all gradings increases with the D_5_ to D_4_ ratio. This is because, under these theoretical mortar conditions, regardless of the D_5_ to D_4_ ratio, the overall mortar quality is insufficient to fill the voids of the coarse aggregate. Hence, the strength fails to reach its peak value. When the mortar increment is 6%, the *R*_T_ of each grading displays a trend of first increasing and then decreasing, with the *R*_T_ peak emerging at a D_5_ to D_4_ ratio of 3:1. This implies that for a 6% mortar increment, a D_5_:D_4_ ratio of 3:1 leads to an adequate amount of D_5_ aggregate coated with asphalt to fill the coarse aggregate voids. A further increase in the D_5_ to D_4_ ratio diminishes the cohesion and internal friction of the mixture, thus causing a decrease in strength. When the mortar increment is 3%, it lies between the two extremes, and only within the T grading does a strength peak appear.

### 4.4. Range of Strong Interlocking Skeleton Dense Gradation

Based on the above research, considering both filling methods, the optimal physical properties and mechanical strength are achieved with T grading and a mortar content of the theoretical amount plus 3%. For the combined filling of asphalt mortar and D_4_, the optimal D_5_:D_4_ ratio is 3:1. Therefore, a comparison of the two filling methods across various indicators is shown in Table 18.

As shown in Table 18, the void ratios are all within the standard range. Comparatively, when D_5_ asphalt mortar is combined with D_4_ filling, the physical properties and mechanical strength are superior to those of D_5_ asphalt mortar independent filling. During the experiment, when the D_5_ to D_4_ ratio was 3:1, the specific grading is shown in Table 19. Based on Table 19 and practical engineering considerations, the strong interlocking skeleton dense gradation range for LSAM-50 is proposed, as shown in Table 20.

### 4.5. Verification of Road Performance

#### 4.5.1. High-Temperature Performance

To evaluate the high-temperature performance of asphalt mixtures through the rutting test, our research group previously determined that the rutting plate thickness for LSAM-50 mixtures should be 18 cm. The test conditions are shown in Table 21, and the specimen molding and testing are illustrated in Figure 16. A comparison of the high-temperature performance of asphalt-treated permeable base (ATB-30) and LSAM-50 was conducted under the same test conditions. The gradation of ATB-30 is presented in Table 22 with an asphalt-aggregate ratio of 4.0%.

The high-temperature rutting test results for LSAM-50 and ATB-30 mixtures are shown in Table 23.

As shown in Table 23, the average dynamic stability of the LSAM-50 mixture is approximately five times that of the ATB-30 mixture. After rolling, the LSAM-50 large thickness rutting plate exhibited no obvious wheel tracks, whereas the ATB-30 large thickness rutting plate displayed more distinct wheel tracks. The higher interlocking force of the skeleton structure gives the LSAM-50 mixture superior high-temperature performance.

#### 4.5.2. Low-Temperature Performance

The low-temperature performance of the LSAM-50 mixture was evaluated using the low-temperature bending test and the SCB (Semi-Circular Bending) low-temperature test. Our research group determined the small beam specimen dimensions to be 60 mm × 65 mm × 280 mm with a test span of 260 mm, and the semi-circular bending specimen dimensions to be 200 mm × 60 mm (thickness) with a test span of 160 mm. The test temperature was set at −15 °C, with a loading rate of 50 mm/min. The surface and cross-section of the two types of mixture specimens are shown in Figure 17. The low-temperature performance test results of the LSAM-50 and ATB-30 mixtures are presented in Table 24.

As shown in Table 24, compared to ATB-30, the LSAM-50 mixture exhibits a 3% increase in low-temperature bending strain and a 47% increase in low-temperature SCB bending strength. This is due to the higher interlocking force of the skeleton structure in the LSAM-50 mixture. However, the asphalt content and properties significantly impact the low-temperature performance of asphalt mixtures, which is why the low-temperature bending strength of the LSAM-50 mixture does not show a significant improvement over the ATB-30 mixture.

#### 4.5.3. Water Stability Performance

The water stability of the LSAM-50 mixture was evaluated using the residual SCB bending strength. The water stability test results of the LSAM-50 and ATB-30 mixtures are shown in Table 25.

As shown in Table 25, the residual SCB strength of both LSAM-50 and ATB-30 mixtures is close to 90%, indicating that the water stability performance of LSAM-50 meets the usage requirements.

## 5. Discussion

P.S. Kandhal studied and calibrated the parameters of the Great Marshall test, which was formulated as the ASTMD 5581 design standard in the 1960s, and then AI improved the design method of the Great Marshall; the Pennsylvania Department of Transportation studied the relationship between the physical and mechanical indicators of the 4-inch standard Marshall specimen and the 6-inch Large Marshall specimen, and suggested that the nominal maximum particle size of the large-particle asphalt mixture should not exceed 37.5 mm, and the large Marshall design method was adopted [32]; AAMAS research pointed out that the rotary compaction design method of the 37.5–53 mm mixture was better than the large Marshall design method. By the 1990s, the design method of the LSAM in NCHRP Report 386 did not specify its control points and restricted areas, emphasizing the control of porosity and powder-to-glue ratio [33]; the Texas Transportation Society studied and gave the dense grading and water permeability design methods for two types of large-grain asphalt mixtures; the South African Asphalt Association has proposed LSAM design guidelines with a particle size of no more than 37.5 mm. Thus, the grading design method and mixture mix ratio design method mentioned in the literature are all aimed at the asphalt mixture with the nominal maximum particle size of aggregate less than or equal to 37.5 mm, but the mixture with the nominal maximum particle size of 53 mm is not involved.

In this study, based on particle accumulation theory, filling theory, and i-method design theory, the optimal composition of coarse aggregate and the optimal ratio of fine aggregate were studied, respectively. Based on the principle of fully filling the gap of coarse aggregate skeleton with fine aggregate, LSAM-50 mixed VVTM specimens with different mortar increments, different coarse aggregate gradations, and different D4 (4.75~9.5 mm aggregate) proportions were formed. The optimal gradation of mechanical properties and the principle a 95% strength guarantee rate were recommended.

LSAM-50 pavement plays a great reference and guide role in the development of long-life pavement in the world. The application of this research result is expected to alleviate the frequent early diseases of asphalt pavement and improve the durability of asphalt pavement. In addition, pavement structure and material innovation are of great significance to improve the corresponding technical specifications and promote the progress of asphalt pavement technology. Its main advantage is that

①The thickness of the surface layer and base material of the flexible pavement is lower than that of the semi-rigid base pavement, which can save the fixed construction costs required for bonding materials, stone, and stone crushing;②Large particle size asphalt mixture single paving, rolling thickness, saving construction costs;③The expected service life of flexible pavement is longer, the expected design life is large, the number of medium repairs is small, the number and time of closed traffic are small, and the indirect economic and social benefits are good;④It can avoid the environmental pollution and solid waste pollution caused by the production of inorganic binders and has good environmental benefits.

The application of SAM-50 flexible base can effectively alleviate the widespread cracking problem of semi-rigid base asphalt pavement, greatly prolong the service life of asphalt pavement, delay the time for large and medium repairs, and reduce vehicle operating costs; the construction and maintenance of pavement not only consumes a lot of funds and resources but also generates a lot of carbon emissions, and frequent maintenance seriously affects the traffic capacity and efficiency of the road network. The application of the LSAM-50 flexible base can significantly reduce the project cost, maintain a good public image of the industry, and have significant social benefits.

However, there are still many shortcomings in this experiment. In this paper, a series of numerical simulations of super-large particle size asphalt mixture LSAM-50 using PFC 5.0 all use ideal spherical particles, and the influence of the irregular shape of the particles on the clearance rate and mechanical properties of the mineral is difficult to reflect. It is recommended that the subsequent use of PFC 6.0 to generate random irregular particles can further improve the simulation accuracy and relevance to indoor tests. At present, although some progress has been made in the development of mineral grading of the LSAM-50 mixture, there are still many limitations, and there are certain limitations in the selection of raw materials. The LSAM-50 mixture usually requires coarse aggregates of larger size; however, suitable sources of large particle size aggregates are not always stable and reliable. Some regions may lack high-quality large-particle stone resources, which makes it necessary to rely on long-distance transportation or the use of inferior quality alternative materials in the process of mineral grading development, which increases the difficulty of cost and quality control. Furthermore, the test and testing methods are relatively lagging behind. The performance testing of large-particle mixtures requires specialized test equipment and methods, but current tests and testing methods often cannot fully meet the requirements. For example, for the strength and durability testing of large-particle mixtures, existing test methods may not accurately simulate the stress state and environmental conditions in actual engineering, resulting in deviations between the test results and actual performance. The performance of the LSAM-50 mixture needs to be verified by a large number of physical projects.

## 6. Conclusions

The LSAM-50 asphalt mixture featuring a strong interlocking skeleton dense gradation has been developed, and its road performance has been verified. The following conclusions have been drawn:(1)For the single-grade coarse aggregate (D_1_) within the range of 37.5–53 mm, the minimum void ratio is 49.2%. In the case of the two-grade mixture composed of D_1_ and the 19–37.5 mm aggregate (D_2_), the minimum void ratio is 45.1%, and the optimal ratio of D_1_:D_2_ is 7:3. For the three-grade mixture consisting of D_1_, D_2_, and the 9.5–19 mm aggregate (D_3_), the minimum void ratio is 44.3%, with the optimal D_1_:D_2_:D_3_ ratio being 5:2:3.(2)The i-method was employed to investigate the impact of fine aggregate gradation on the strength of asphalt mortar. Based on the principle of optimal mechanical strength, the recommended fine aggregate gradation is obtained when i = 0.75.(3)When D_5_ asphalt mortar is used for independent filling, both the compressive strength (*R*_c_) and bulk density of the specimen initially increase and subsequently decrease as the mortar content rises, reaching their peaks when the mortar content is 3% above the theoretical amount. The void ratio decreases linearly with the increase in mortar content, while the tensile strength (*R*_T_) increases linearly with the increase in mortar content. An increase in coarse aggregate content leads to an increase in bulk density, a decrease in void ratio, and an enhancement in the mechanical strength of the specimen.(4)When D_5_ asphalt mortar combined with D_4_ is used for filling, the bulk density of the specimen increases and the void ratio decreases with the increase in mortar content. Rc first increases and then decreases, with most of the peak values emerging when the mortar content is increased by 3%. As the coarse aggregate content increases, the bulk density of the specimen decreases, the void ratio first increases and then decreases, and the mechanical strength increases with a higher content of coarse aggregate. When the ratio of D_5_:D_4_ is 3:1, the physical and mechanical test indicators of the specimen are at their optimal state.

It should be emphasized that these findings are derived from laboratory tests. Further validation through practical road construction projects will be carried out to confirm these macro-level experimental results.

## Figures and Tables

**Figure 1 materials-17-05712-f001:**
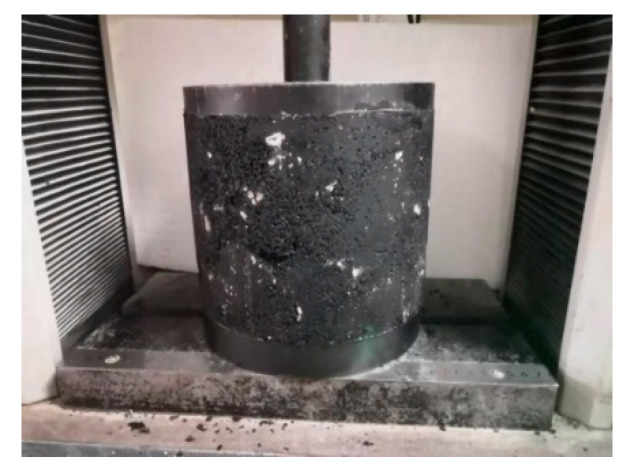
Unconfined compressive strength test.

**Figure 2 materials-17-05712-f002:**
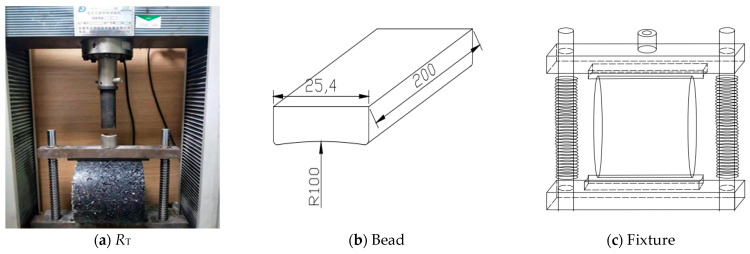
Splitting test (unit: mm).

**Figure 3 materials-17-05712-f003:**
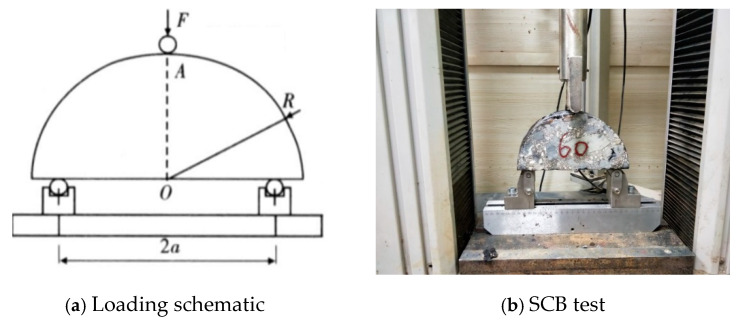
Semi-circular bending strength test.

**Figure 4 materials-17-05712-f004:**
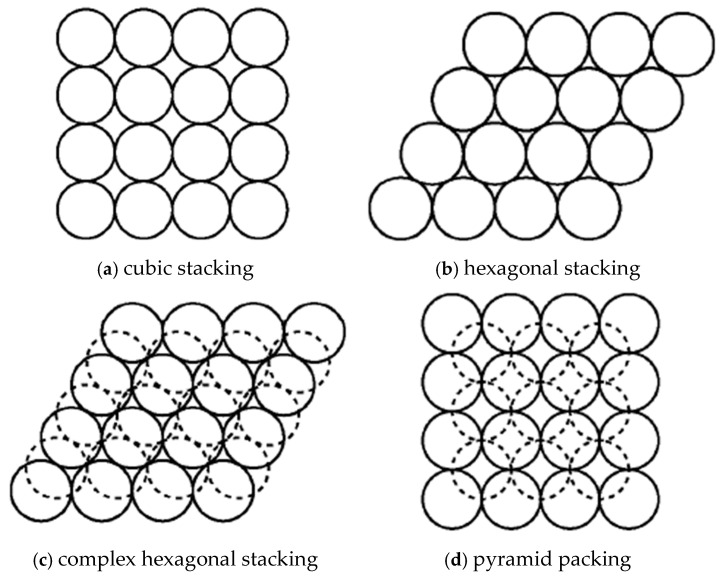
Typical particle accumulation method.

**Figure 5 materials-17-05712-f005:**
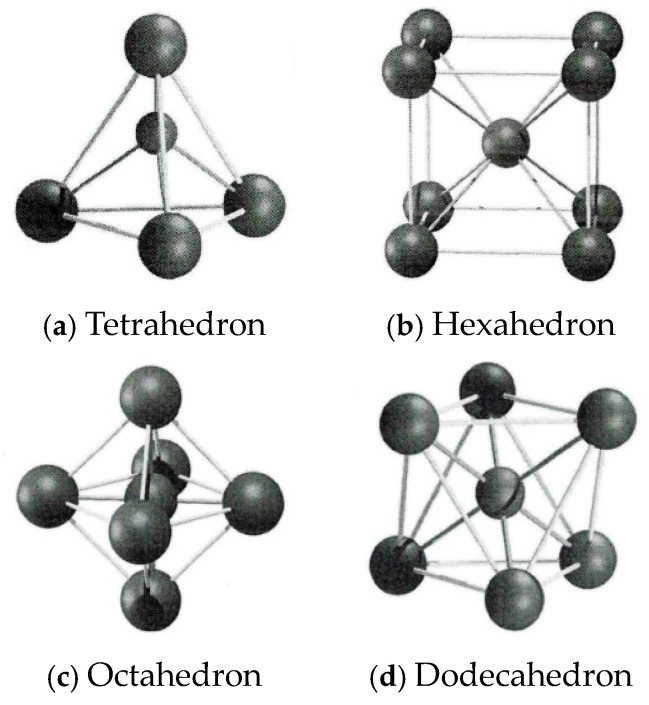
Ordered packing ball-and-stick models of spherical particles.

**Figure 6 materials-17-05712-f006:**
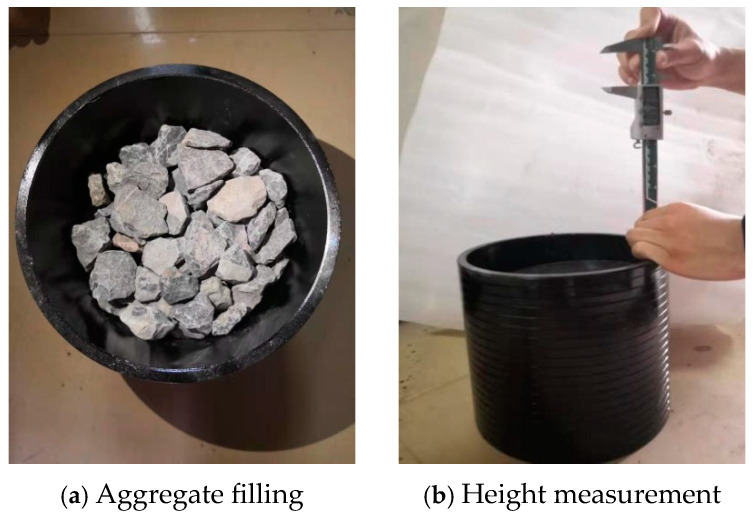
Schematic diagram of filling and measurement.

**Figure 7 materials-17-05712-f007:**
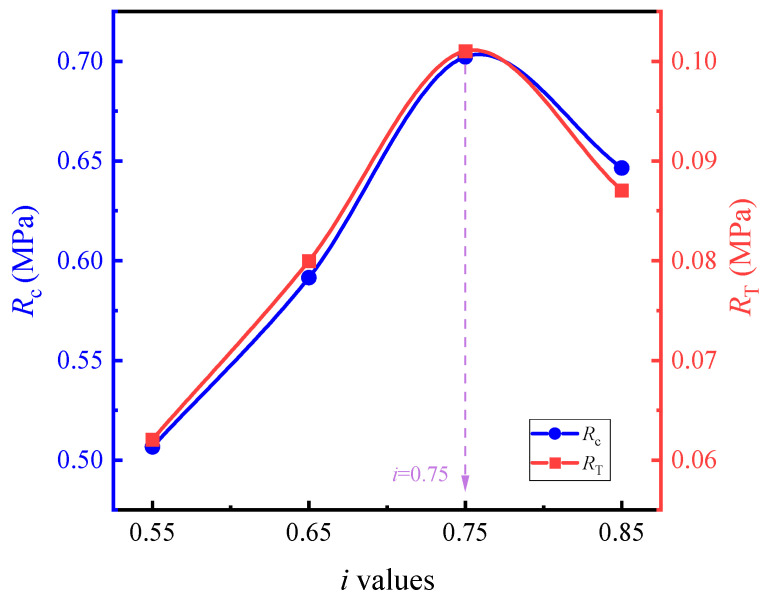
The effects of different *i* values on the mechanical strength of asphalt mortar.

**Figure 8 materials-17-05712-f008:**
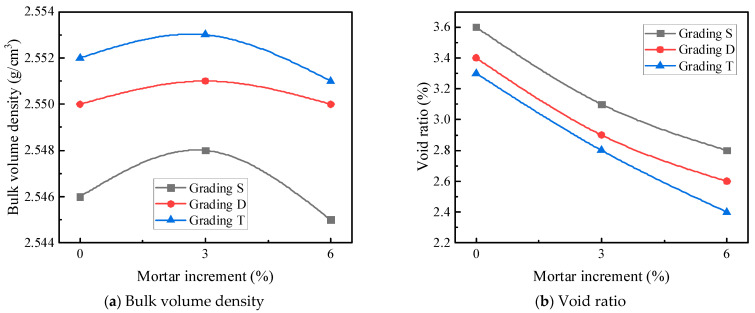
The effects of asphalt mortar content on the physical properties of LSAM-50 specimens.

**Figure 9 materials-17-05712-f009:**
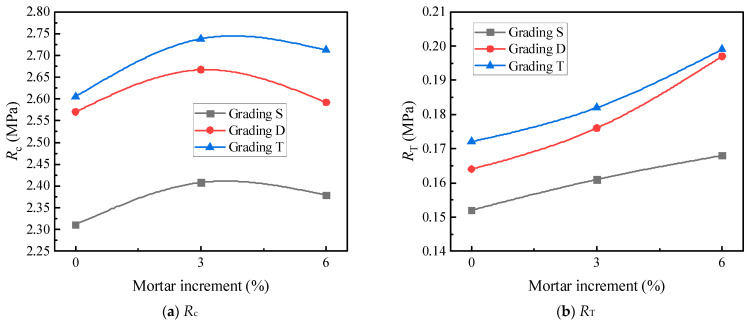
The effects of asphalt mortar content on the mechanical strength of LSAM-50 specimens.

**Figure 10 materials-17-05712-f010:**
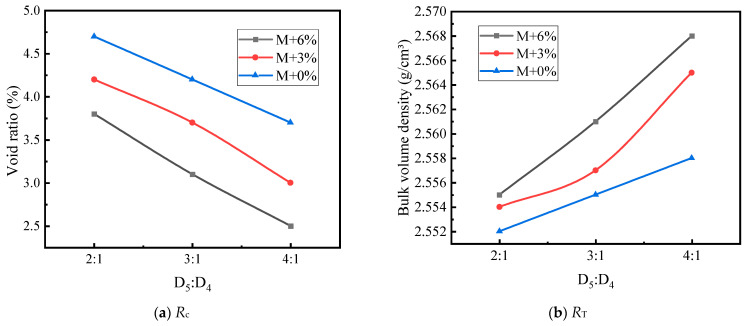
The trend of mechanical strength variation with the D_5_ to D_4_ ratio under S grading.

**Figure 11 materials-17-05712-f011:**
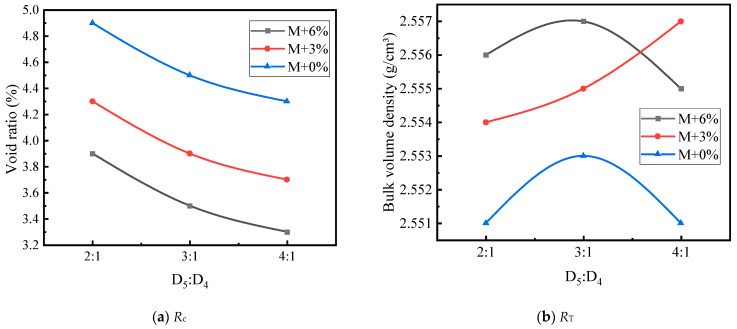
The trend of mechanical strength variation with the D_5_ to D_4_ ratio under S grading.

**Figure 12 materials-17-05712-f012:**
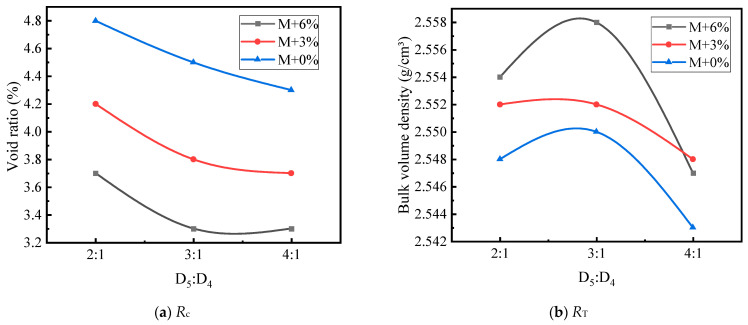
The trend of mechanical strength variation with the D_5_ to D_4_ ratio under S grading.

**Figure 13 materials-17-05712-f013:**
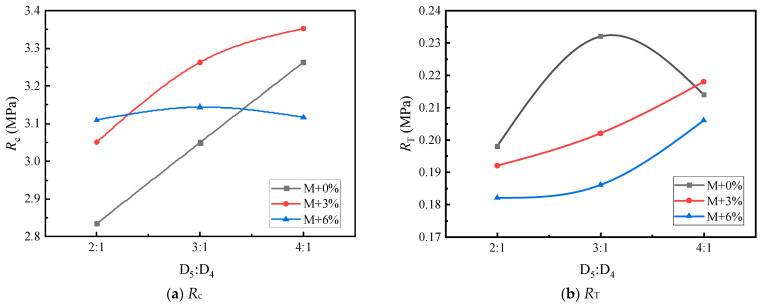
The trend of mechanical strength variation with the D_5_ to D_4_ ratio under S grading.

**Figure 14 materials-17-05712-f014:**
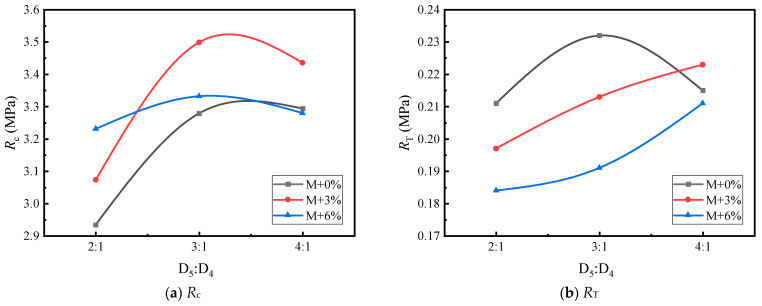
The trend of mechanical strength variation with the D_5_ to D_4_ ratio under D grading.

**Figure 15 materials-17-05712-f015:**
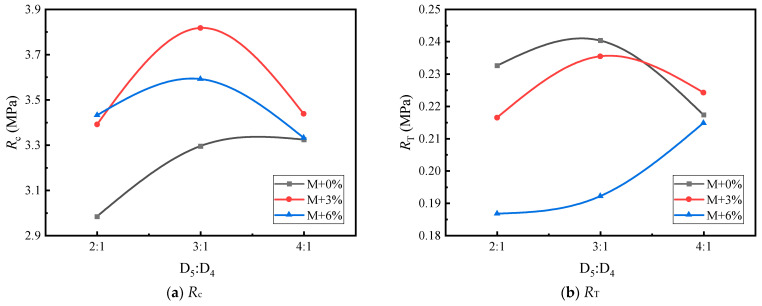
The trend of mechanical strength variation with the D_5_ to D_4_ ratio under T grading: (**a**,**b**).

**Figure 16 materials-17-05712-f016:**
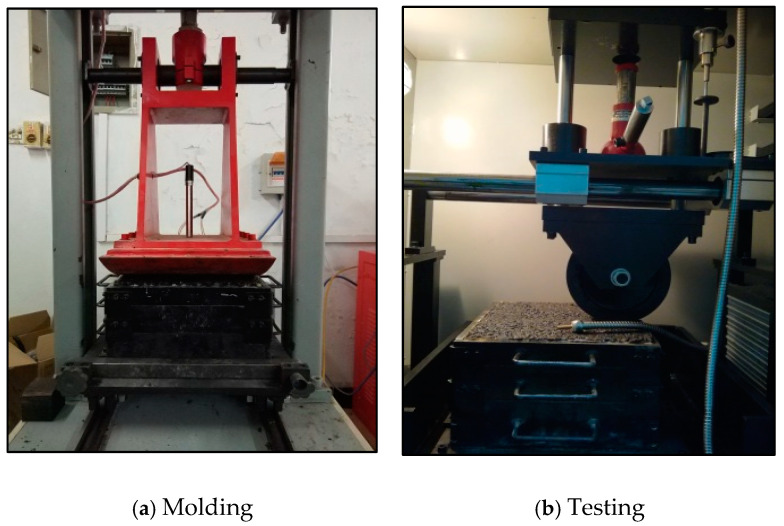
Schematic drawing of molding and testing for large thickness rutting plates.

**Figure 17 materials-17-05712-f017:**
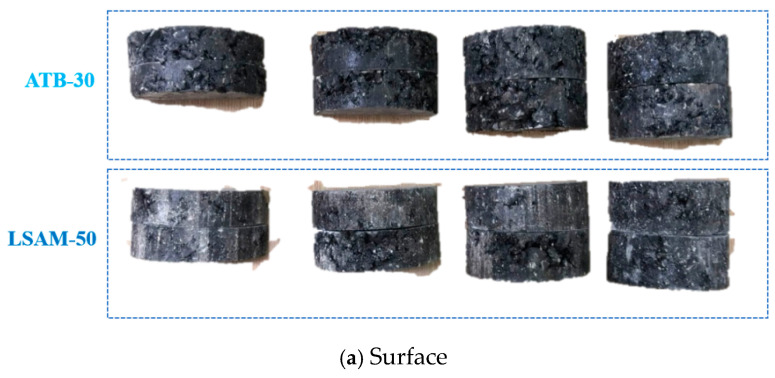
Surface and cross-section of semi-circular bending specimens.

**Table 1 materials-17-05712-t001:** Technical properties of Esso A-70 base asphalt and SBS (I-C)-modified asphalt.

Items		Esso A-70
Penetration (25 °C, 0.01 mm)		68
Ductility (5 cm/min,10 °C, cm)		38
Softening point (°C)		47.3
Density (15 °C, g/cm^3^)		1.035
Rotating film aging test (163 °C, 85 min)	Mass lose (%)	0.03
	Penetration ratio (25 °C, %)	63
	Ductility (5 cm/min,10 °C, cm)	7

**Table 2 materials-17-05712-t002:** Technical properties of aggregates.

Item	Aggregate Size (mm)					
	37.5–53	19–37.5	9.5–19	4.75–9.5	2.36–4.75	0–2.36
Apparent relative density	2.819	2.779	2.755	2.721	2.728	2.728
Flakiness content (%)	2.2	7.9	7.3	13.1	—	-
Water absorption (%)	0.37	0.62	0.98	1.73	0.75	-
rushed value (%)	—	18.5	—	—	—	-

**Table 3 materials-17-05712-t003:** Working parameters of the VVTE.

Work Frequency (Hz)	Work Amplitude (mm)	Upper-System Weight (kg)	Lower-System Weight (kg)	Vibration Time (s)
40	1.2	122	180	90

**Table 4 materials-17-05712-t004:** Typical stacking mode and clearance ratio.

Stacking Method	Coordination Number	Theoretical Gap Rate (%)
cubic stacking	6	47.64
hexagonal stacking	8	39.55
complex hexagonal stacking	10	30.19
pyramid packing	12	25.95

**Table 5 materials-17-05712-t005:** Gap ratio under different distribution modulus of stacking model.

**Distributed Modulus**	0.33	0.35	0.37	0.39	0.41
**Clearance Ratio (%)**	43.17	42.22	41.33	40.43	39.53

**Table 6 materials-17-05712-t006:** Void ratio of single-size aggregate skeleton.

Aggregate Particle Size Range	D_1_	D_2_	D_3_	D_4_
Void Ratio (%)	49.2	48.9	47.3	46.6

**Table 7 materials-17-05712-t007:** Void ratio of the coarse aggregate skeleton filled with D_1_ and D_2_.

**Mass Ratio of D_1_ to D_2_**	64:36	67:33	70:30	73:27	76:24
**Void Ratio (%)**	49.1	48.0	45.1	45.6	47.2

**Table 8 materials-17-05712-t008:** Packing density and void ratio of (D_1_ + D_2_) filled with D_3_.

Test Index	Test Results for the Following Mass Ratios of (D_1_ + D_2_) to D_3_							
	90:10	85:15	80:20	75:25	72:28	70:30	68:32	65:35
Packing density (g/cm^3^)	1.471	1.489	1.522	1.545	1.551	1.557	1.537	1.523
Void ratio (%)	46.6	46.2	45.9	44.6	44.4	44.3	44.8	45.4

**Table 9 materials-17-05712-t009:** The fine aggregate gradations corresponding to different *i* values.

*i* Values	The Mass Percentages (%) Passing Through the Following Sieve Sizes (mm)						
	4.75	2.36	1.18	0.6	0.3	0.15	0.075
0.55	100	55	30.3	16.6	9.2	5	2.8
0.65	100	65	42.3	27.5	17.9	11.6	7.5
0.75	100	75	56	42.2	31.6	23.7	17.8
0.85	100	85	72	61.6	52.4	44.5	37.8

**Table 10 materials-17-05712-t010:** The mechanical strength of asphalt mortar corresponding to different *i* values.

Mechanical Strength	The *R*_c_ and *R*_T_ of Asphalt Mortar Corresponding to the Following *i* Values (MPa)			
	0.55	0.65	0.75	0.85
*R* _c_	0.507	0.592	0.702	0.647
*R* _T_	0.062	0.080	0.101	0.087

**Table 11 materials-17-05712-t011:** Gradation of fine aggregate.

**Sieve Sizes (mm)**	4.75	2.36	1.18	0.6	0.3	0.15	0.075
**Passing Rate (%)**	100	75	56	42.2	31.6	23.7	17.8

**Table 12 materials-17-05712-t012:** The theoretical amounts of asphalt mortar for each grading type when D_5_ is used to fill independently.

Grading Types	S	D	T
Theoretical amounts of asphalt mortar (%)	45.3	41.5	41.0

**Table 13 materials-17-05712-t013:** The theoretical amounts of asphalt mortar for each grading type when D_5_ and D_4_ are used to fill jointly.

Grading Types	The Theoretical Amounts of Asphalt Mortar for Each Grading Type Under the Following D_5_:D_4_ Ratios		
	2:1	3:1	4:1
S	31.4	34.9	36.9
D	28.7	31.9	33.8
T	28.4	32.1	33.9

**Table 14 materials-17-05712-t014:** Bulk volume density test results for D_5_ and D_4_ joint filling.

Grading Types	Asphalt Mortar Content (%)	Bulk Volume Density (g/cm^3^) at the Following D_5_ to D_4_ Ratios		
		2:1	3:1	4:1
S	Theoretical mortar content (M + 0%)	2.552	2.555	2.558
	Theoretical mortar content + 3% (M + 3%)	2.554	2.557	2.565
	Theoretical mortar content + 6% (M + 6%)	2.555	2.561	2.568
D	Theoretical mortar content (M + 0%)	2.551	2.553	2.551
	Theoretical mortar content + 3% (M + 3%)	2.554	2.555	2.557
	Theoretical mortar content + 6% (M + 6%)	2.556	2.557	2.555
T	Theoretical mortar content (M + 0%)	2.548	2.550	2.543
	Theoretical mortar content + 3% (M + 3%)	2.552	2.552	2.548
	Theoretical mortar content + 6% (M + 6%)	2.554	2.558	2.547

**Table 15 materials-17-05712-t015:** Void ratio test results for D_5_ and D_4_ joint filling.

Grading Types	Asphalt Mortar Content (%)	Void Ratio (%) at the Following D_5_ to D_4_ Ratios		
		2:1	3:1	4:1
S	Theoretical mortar content (M + 0%)	4.7	4.2	3.7
	Theoretical mortar content + 3% (M + 3%)	4.2	3.7	3.0
	Theoretical mortar content + 6% (M + 6%)	3.8	3.1	2.5
D	Theoretical mortar content (M + 0%)	4.9	4.5	4.3
	Theoretical mortar content + 3% (M + 3%)	4.3	3.9	3.7
	Theoretical mortar content + 6% (M + 6%)	3.9	3.5	3.3
T	Theoretical mortar content (M + 0%)	4.8	4.5	4.3
	Theoretical mortar content + 3% (M + 3%)	4.2	3.8	3.7
	Theoretical mortar content + 6% (M + 6%)	3.7	3.3	3.3

**Table 16 materials-17-05712-t016:** *R*_c_ test results for D_5_ and D_4_ joint filling.

Grading Types	Asphalt Mortar Content (%)	*R*_c_ (MPa) at the Following D_5_ to D_4_ Ratios		
		2:1	3:1	4:1
S	Theoretical mortar content (M + 0%)	2.835	3.050	3.263
	Theoretical mortar content + 3% (M + 3%)	3.051	3.263	3.352
	Theoretical mortar content + 6% (M + 6%)	3.110	3.144	3.117
D	Theoretical mortar content (M + 0%)	2.934	3.279	3.294
	Theoretical mortar content + 3% (M + 3%)	3.073	3.499	3.436
	Theoretical mortar content + 6% (M + 6%)	3.231	3.332	3.280
T	Theoretical mortar content (M + 0%)	2.984	3.295	3.324
	Theoretical mortar content + 3% (M + 3%)	3.391	3.818	3.438
	Theoretical mortar content + 6% (M + 6%)	3.432	3.592	3.332

**Table 17 materials-17-05712-t017:** *R*_T_ test results for D_5_ and D_4_ joint filling.

Grading Types	Asphalt Mortar Content (%)	*R*_T_ (MPa) at the Following D_5_ to D_4_ Ratios		
		2:1	3:1	4:1
S	Theoretical mortar content (M + 0%)	0.182	0.186	0.206
	Theoretical mortar content + 3% (M + 3%)	0.192	0.202	0.218
	Theoretical mortar content + 6% (M + 6%)	0.198	0.232	0.214
D	Theoretical mortar content (M + 0%)	0.184	0.191	0.211
	Theoretical mortar content + 3% (M + 3%)	0.197	0.213	0.223
	Theoretical mortar content + 6% (M + 6%)	0.211	0.232	0.215
T	Theoretical mortar content (M + 0%)	0.187	0.192	0.215
	Theoretical mortar content + 3% (M + 3%)	0.216	0.235	0.224
	Theoretical mortar content + 6% (M + 6%)	0.233	0.240	0.217

**Table 18 materials-17-05712-t018:** Comparison of physical properties and mechanical strength under two filling methods.

Filling Method	Bulk Volume Density (g/cm^3^)	Void Ratio (%)	*R*_c_ (MPa)	*R*_T_ (MPa)
D_5_ asphalt mortar independent filling	2.553	2.8	2.58	0.171
D_5_ asphalt mortar combined with D_4_ filling	2.555	3.8	3.72	0.204

**Table 19 materials-17-05712-t019:** Gradation for each asphalt mortar content at a D5 to D4 ratio of 3:1.

Grading Types	Asphalt Mortar Content (%)	Percentage Passing Through the Following Sieve Sizes (mm)										
		53	37.5	19	9.5	4.75	2.36	1.18	0.6	0.3	0.15	0.075
S	Theoretical mortar content (M + 0%)	100	43.8	43.8	43.8	33.0	24.8	18.5	13.9	10.4	7.8	5.9
	Theoretical mortar content + 3% (M + 3%)	100	46.7	46.7	46.7	36.0	27.0	20.1	15.2	11.4	8.5	6.4
	Theoretical mortar content + 6% (M + 6%)	100	46.9	46.9	46.9	32.6	24.4	18.2	13.7	10.3	7.7	5.8
D	Theoretical mortar content (M + 0%)	100	40	40	40	30.2	22.6	16.9	12.7	9.5	7.2	5.4
	Theoretical mortar content + 3% (M + 3%)	100	59.7	42.9	42.9	33.1	24.8	18.5	14.0	10.5	7.8	5.9
	Theoretical mortar content + 6% (M + 6%)	100	59.8	43.1	43.1	30.0	22.5	16.8	12.7	9.5	7.1	5.3
T	Theoretical mortar content (M + 0%)	100	69.7	57.1	39.5	30.3	22.8	17.0	12.8	9.6	7.2	5.4
	Theoretical mortar content + 3% (M + 3%)	100	71.2	59.2	42.4	33.3	24.9	18.6	14.0	10.5	7.9	5.9
	Theoretical mortar content + 6% (M + 6%)	100	71.3	59.3	42.6	29.7	22.2	16.6	12.5	9.4	7.0	5.3

**Table 20 materials-17-05712-t020:** Strong interlocking skeleton dense gradation range for LSAM-50.

**Sieve Size (mm)**	63	53	37.5	19	9.5	4.75	2.36	1.18	0.6	0.3	0.15	0.075
**Passing Rate (%)**	100	90–100	65–75	55–65	37–47	30–38	22–30	14–22	10–18	7–13	5–10	3–6

**Table 21 materials-17-05712-t021:** Molding and testing conditions for LSAM-50 large thickness rutting plate specimens.

Specimen Thickness (cm)	Number of Rolling Passes	Test Gradation	Asphalt-Aggregate Ratio (%)	Heat Preservation Time (h)	Parallel Specimens (Groups)
18	Bidirectional 46	Strong interlocking skeleton dense gradation	2.7	8	6

**Table 22 materials-17-05712-t022:** Gradation of ATB-30.

**Sieve Size (mm)**	37.5	31.5	26.5	19.0	16.0	13.2	9.5	4.75	2.36	1.18	0.6	0.3	0.15	0.075
**Passing Rate (%)**	100.0	95.0	80.0	62.5	55.0	49.5	41.0	30.0	23.5	17.5	13.0	9.5	6.5	4.0

**Table 23 materials-17-05712-t023:** High-temperature performance test results of LSAM-50 and ATB-30.

Mixture Type	Dynamic Stability (cycles/mm)								Average Value (cycles/mm)
LSAM-50	13,343	12,726	14,991	14,007	15,075	12,896	13,272	14,569	13,860
ATB-30	2732	3222	2440	2676	3225	2263	2625	2761	2743

**Table 24 materials-17-05712-t024:** Low-temperature performance test results of LSAM-50 and ATB-30.

Test Indicators	Mixture Type	Measured Low-Temperature Performance Values of Parallel Specimens			Average Value	Relative Value
Bending strain (με)	LSAM-50	3390	3368	3396	3385	1.032
	ATB-30	3265	3272	3303	3280	1
SCB strength (MPa)	LSAM-50	10.87	10.55	10.56	10.66	1.470
	ATB-30	7.31	7.19	7.18	7.23	1

**Table 25 materials-17-05712-t025:** Water stability performance test results of LSAM-50 and ATB-30.

Mixture Type	Residual SCB Strength Test Results of Parallel Specimens (%)			Average Value (%)
LSAM-50	90.8	90.2	89.4	90.1
ATB-30	87.7	89.6	89.7	89.0

## Data Availability

The data used to support the findings of this study are included within the article.

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
