# Peer review of "Development and Road Performance Verification of Aggregate Gradation for Large Stone Asphalt Mixture"

_materials, 2024, doi:10.3390/ma17235712_

Round 1
Reviewer 1 Report
Comments and Suggestions for Authors
This was a well-studied paper with enough experimental testing, although pretty conventional in nature. There is a need to clearly highlight the originality of the studied subject. Please also insert a methodological framework with the experimental steps followed. Finally, what is the practical contribution of this paper? Is to improve and optimize mix design for heavy duty pavements? Please reconsider and explain the target output for routing practice.
Author Response
Comment :what is the practical contribution of this paper? Is to improve and optimize mix design for heavy duty pavements? Please reconsider and explain the target output for routing practice
Response:Thanks for pointing this out. We agree with this comment. The actual contribution of this paper is to determine the gradation of coarse aggregate through indoor filling test and numerical simulation based on the principle of optimal skeleton embedding force; through indoor test based on the i method, determine the gradation of fine aggregate based on the principle of optimal mechanical strength of asphalt mortar specimens; through numerical test and indoor test, study the influence of the ratio of coarse aggregate to asphalt mortar on the physical and mechanical properties of LSAM-50, and propose the dense gradation of LSAM-50 mixture strong embedding and extrusion skeleton. Based on particle accumulation theory, filling theory and i-method design theory, the optimal composition of coarse aggregate and the optimal ratio of fine aggregate were studied respectively. Based on the principle of fully filling the gap of coarse aggregate skeleton with fine aggregate, LSAM-50 mixed VVTM specimens with different mortar increments, different coarse aggregate gradation and different D4 (4.75~ 9.5mm aggregate) proportions were formed. The optimal gradation of mechanical properties was recommended and the principle of 95% strength guarantee rate was optimized.

Reviewer 2 Report
Comments and Suggestions for Authors
The article is of interest, considering the scale that road transport has acquired and the increasingly large dimensions that move on asphalt roads.
However, when reading this article I found that there are some similarities with previously written articles. Therefore, I specify that authors must mention the bibliographical sources and published articles in which similar information is found (figures, images, calculation relationships, tables and expressions), even if they belong to one or more authors. In this sense, I mention the part of the reviewed article 2.1.1, 2.1.2, 2.2, 2.3 up to figure 2 (page 6). This portion can be said to be taken from the article ”Jiang, Y.; Cai, M.; Li, S.; Zhang, Y.; Yi, Y.; Su, H.; Bai, C. Design of Volume Parameters of Large-Particle-Size Asphalt Mixture Based on the Vertical Vibration Compaction Method. Appl. Sci. 2024, 14, 6983. https://doi.org/10.3390/app14166983”
Other recommendations refer to the following aspects:
- The authors should better justify the use of spherical particles in the modeling. These deviate considerably from the real model. Also, the interest in the Hexahedron model should be better justified. The real shape of the aggregate particles may change the option.
- The writing is sloppy and inaccurate. On page 13, the reference to figure 7 is erroneous (figure 6 should be).
- The results in table 8 should be justified. In the text, the explanations are minor.
- The representations in figure 7 are insufficiently explained in the text. Consequently, the reader has doubts about the representations. Ditto in figure 8.
- Tables 12-15 and figures 9-14 require clearer explanations in the text.
- Figure 15 b is taken from the article “Jiang, Y.; Cai, M.; Li, S.; Zhang, Y.; Yi, Y.; Su, H.; Bai, C. Design of Volume Parameters of Large-Particle-Size Asphalt Mixture Based on the Vertical Vibration Compaction Method. Appl. Sci. 2024, 14, 6983. https://doi.org/10.3390/app14166983”
- The name of Figure 15 is inappropriate. It is not a diagram
It is recommended that, throughout the paper, the results obtained be compared with results obtained and specified in other similar papers.
The conclusions should be developed further and the original contributions noted, by comparison with similar studies in the researched field.
The English language should be corrected, for an easier understanding of the expression.
Comments on the Quality of English LanguageThe English language should be corrected, for an easier understanding of the expression.
Author Response
Comment 1:The authors should better justify the use of spherical particles in the modeling. These deviate considerably from the real model. Also, the interest in the Hexahedron model should be better justified. The real shape of the aggregate particles may change the option.
Response 1:Thanks for pointing this out. We agree with this comment. Therefore, We have provided supplementary explanations for this part anew in the revised draft and introduced the theory in detail.
Comment 2:The writing is sloppy and inaccurate. On page 13, the reference to figure 7 is erroneous (figure 6 should be).
Response 1:Thanks for pointing this out. We agree with this comment. Therefore, We have already made corrections to the charts and diagrams in this part.
Comment 3: - The results in table 8 should be justified. In the text, the explanations are minor.
Response 3:Thanks for pointing this out. We agree with this comment. Therefore, In the revised draft, we have provided supplementary explanations for this part to improve its integrity.
Comment 4: - he representations in figure 7 are insufficiently explained in the text. Consequently, the reader has doubts about the representations. Ditto in figure 8.
Response 4:Thanks for pointing this out. We agree with this comment. Therefore, In the revised draft, we have provided supplementary explanations for this part to improve its integrity.
Comment 5: Tables 12-15 and figures 9-14 require clearer explanations in the text.the reader has doubts about the representations. Ditto in figure 8.
Response 5:Thanks for pointing this out. We agree with this comment. Therefore, We have re - sorted and re - analyzed this part in the revised manuscript to ensure readability for readers and the integrity of the content.
Comment 6: The name of Figure 15 is inappropriate. It is not a diagram
Response 6:Thanks for pointing this out. We agree with this comment. Therefore, We have replaced the title of this part with (Schematic drawing of molding and testing for large thickness rutting plates).
Comment 7: It is recommended that, throughout the paper, the results obtained be compared with results obtained and specified in other similar papers.
The conclusions should be developed further and the original contributions noted, by comparison with similar studies in the researched field.
Response 7:Thanks for pointing this out. We agree with this comment. Therefore, In the revised manuscript, we have introduced a new section named "Discussion". In this part, we have restated the previously completed research within the field, emphasized and highlighted the contributions of this study, and also underlined the current research limitations and possible future development scenarios.

Reviewer 3 Report
Comments and Suggestions for Authors
I have thoroughly reviewed the manuscript titled "A Development of Aggregate Gradation for Large Stone Asphalt Mixture and Verification of Road Performance" and find that it aligns well with the scope of the Materials journal published by MDPI. However, I would like to submit several major revision requests for consideration:
1. In the "Abstract" section, it is important to explicitly state the research hypotheses.
2. The "Introduction" section currently lacks a clear presentation of the research hypotheses at its conclusion. Moreover, these hypotheses should be addressed, either verified or refuted, in the conclusions of the paper. At this juncture, the objective of the study must be articulated with precision, and the needs should be contextualized within an international framework to identify the final beneficiaries.
3. In the "Materials and Methods" section, I recommend referencing international standards, such as ASTM, rather than JTG, to ensure accessibility for a wider audience.
4. The section titled "Mechanism of Strength Formation and Gradation Design Method" requires the completion of references to pertinent literature sources.
5. In the section "Development of a Densely Graded Strong Interlocking Skeleton for LSAM-50," it is advisable to rewrite and condense the content. Currently, research results are presented redundantly in both tables and figures. Additionally, appropriate citations to relevant literature sources should be included.
6. The manuscript would benefit from the inclusion of a "Discussion" section. This section should delineate the mutual influence of the assessed aspects and identify other factors that may impact them. Furthermore, the discussion should be expanded to extend its relevance on a global scale and examine potential industrial applications of the laboratory solution, thereby enhancing the paper's analytical depth.
7. In the "Conclusion" section, the current summary of conducted studies needs to be revised. The conclusions should be succinct and emphasize the key outcome data for increased clarity. Moreover, it is essential to include established hypotheses, theoretical and engineering implications, highlight research limitations, and propose directions for future research.
In summary, the manuscript requires a comprehensive revision, and adherence to the MDPI template is recommended to ensure professional presentation.
Author Response
Comment 1: In the "Abstract" section, it is important to explicitly state the research hypotheses.
Response 1:Thanks for pointing this out. We agree with this comment. Therefore, In the revised manuscript, we have reorganized the abstract part and clarified the necessity of this study and the assumptions of the research objects.
Comment 2: The "Introduction" section currently lacks a clear presentation of the research hypotheses at its conclusion. Moreover, these hypotheses should be addressed, either verified or refuted, in the conclusions of the paper. At this juncture, the objective of the study must be articulated with precision, and the needs should be contextualized within an international framework to identify the final beneficiaries.
Response 2:Thanks for pointing this out. We agree with this comment. Therefore, In the revised draft, we have added a new "Discussion" section, reclarifying the research objectives, the work completed in this study, and the impact of this work within the international framework, including the verification of assumptions.
Comment 3: In the "Materials and Methods" section, I recommend referencing international standards, such as ASTM, rather than JTG, to ensure accessibility for a wider audience.
Response 3:Thanks for pointing this out. We agree with this comment. However, JTG refers to Chinese standards. Currently, most of the research on LSAM - 50 materials is conducted in China. After this research matures, international norms will be gradually referred to.
Comment 4: The section titled "Mechanism of Strength Formation and Gradation Design Method" requires the completion of references to pertinent literature sources
Response 4:Thanks for pointing this out. We agree with this comment. However, This part is the LSAM - 50 design concept independently proposed with reference to the particle packing theory, filling theory, i - method design theory and the design concept of asphalt
Comment 5 In the section "Development of a Densely Graded Strong Interlocking Skeleton for LSAM-50," it is advisable to rewrite and condense the content. Currently, research results are presented redundantly in both tables and figures. Additionally, appropriate citations to relevant literature sources should be included.
Response 5: Thanks for pointing this out. We agree with this comment. Therefore, We have provided supplementary explanations for this part anew in the revised draft and introduced the theory in detail.
Comment 6: The manuscript would benefit from the inclusion of a "Discussion" section. This section should delineate the mutual influence of the assessed aspects and identify other factors that may impact them. Furthermore, the discussion should be expanded to extend its relevance on a global scale and examine potential industrial applications of the laboratory solution, thereby enhancing the paper's analytical depth..
Response 6: Thanks for pointing this out. We agree with this comment. Therefore, we have added a new "Discussion" section, reclarifying the research objectives, the work completed in this study, and the impact of this work within the international framework, including the verification of assumptions.
Comment 7: In the "Conclusion" section, the current summary of conducted studies needs to be revised. The conclusions should be succinct and emphasize the key outcome data for increased clarity. Moreover, it is essential to include established hypotheses, theoretical and engineering implications, highlight research limitations, and propose directions for future research.
Response 7: Thanks for pointing this out. We agree with this comment. Therefore, In the revised manuscript, we have reorganized the conclusion part, clarified the assumptions, theories and research findings, and proposed the future research directions.

Round 2
Reviewer 2 Report
Comments and Suggestions for Authors
I appreciate the authors' efforts in correcting the article. In subchapter 4.1.2, I think that, after the title, a short introductory phrase would be appropriate. The entry is too brutal in the casuistry (it starts directly with point "a")
Author Response
Comment :. In subchapter 4.1.2, after the title, a short introductory phrase would be appropriate. The entry is too brutal in the casuistry (it starts directly with point "a")
Response:Thank you for pointing this out.We agree with that.Thus,We have added the explanation of the basic packing theory and the basic interpretation of the concept of porosity in this part, laying a good foundation for the subsequent packing theory.

Reviewer 3 Report
Comments and Suggestions for Authors
The authors have made considerable enhancements to the manuscript, effectively addressing the majority of my comments and suggestions. They have clarified key points, improved the overall structure, and strengthened the arguments presented. I am confident that these revisions have elevated the quality of the work, making it suitable for publication.
Author Response
Thank you for your affirmation. We will spare no effort to refine the article.